# Risk factors affecting spinal fusion: A meta-analysis of 39 cohort studies

**Shudong Yang[1‡], Beijun Zhou[2‡], Jiaxuan Mo[2], Ruidi He[2], Kunbo Mei[2], Zhi Zeng[2], Gaigai Yang[2], Yuwei Chen[2], Mingjiang Luo**[3]*, **Siliang Tang[4]**\*, **Zhihong Xiao[3]**\*

**1** Department of Orthopedic Trauma, Second Affiliated Hospital, Hengyang Medical School, University of South China, Hengyang City, Hunan Province, China, **2** Hengyang Medical School, University of South China, Hengyang City, Hunan Province, China, **3** Department of Spine Surgery, Lishui Hospital of Wenzhou Medical University, Lishui People's Hospital, Lishui, Zhejiang, China, **4** Department of Spine Surgery, Second Affiliated Hospital, Hengyang Medical School, University of South China, Hengyang City, Hunan Province, China

‡ SY and BZ share co-first authors on this work.
* 498381035@qq.com (ST); 1982622526@qq.com (ML); 35042875@qq.com (ZX)

## Abstract

### Purpose

We performed a meta-analysis to identify risk factors affecting spinal fusion.

### Methods

We systematically searched PubMed, Embase, and the Cochrane Library from inception to January 6, 2023, for articles that report risk factors affecting spinal fusion. The pooled odds ratios (ORs) and 95% confidence intervals (CIs) were estimated using fixed-effects models for each factor for which the interstudy heterogeneity $I^2$ was < 50%, while random-effects models were used when the interstudy heterogeneity $I^2$ was ≥ 50%. Using sample size, Egger's P value, and heterogeneity across studies as criteria, we categorized the quality of evidence from observational studies as high-quality (Class I), moderate-quality (Class II or III), or low-quality (Class IV). Furthermore, the trim-and-fill procedure and leave-one-out protocol were conducted to investigate potential sources of heterogeneity and verify result stability.

### Results

Of the 1,257 citations screened, 39 unique cohort studies comprising 7,145 patients were included in the data synthesis. High-quality (Class I) evidence showed that patients with a smoking habit (OR, 1.57; 95% CI, 1.11 to 2.21) and without the use of bone morphogenetic protein-2 (BMP-2) (OR, 4.42; 95% CI, 3.33 to 5.86) were at higher risk for fusion failure. Moderate-quality (Class II or III) evidence showed that fusion failure was significantly associated with vitamin D deficiency (OR, 2.46; 95% CI, 1.24 to 4.90), diabetes (OR, 3.42; 95% CI, 1.59 to 7.36), allograft (OR, 1.82; 95% CI, 1.11 to 2.96), conventional pedicle screw (CPS) fixation (OR, 4.77; 95% CI, 2.23 to 10.20) and posterolateral fusion (OR, 3.63; 95% CI, 1.25 to 10.49).

**Data Availability Statement:** Study data are available at request from the corresponding author.

**Funding:** The author(s) received no specific funding for this work.

**Competing interests:** The authors have declared that no competing interests exist.

## Conclusions

Conspicuous risk factors affecting spinal fusion include three patient-related risk factors (smoking, vitamin D deficiency, and diabetes) and four surgery-related risk factors (without the use of BMP-2, allograft, CPS fixation, and posterolateral fusion). These findings may help clinicians strengthen awareness for early intervention in patients at high risk of developing fusion failure.

## Introduction

Spinal disease is a common clinical surgical disease, which is usually caused by lesions of the vertebral body and its surrounding soft tissue or spinal canal. Common spinal disorders include spinal degenerative diseases, inflammation, tumors, spinal deformity, and spinal fracture. These diseases can cause pain or neurological dysfunction, thus leading to a significant reduction in the patient's quality of life and ability to work [1] In recent years, the prevalence of spinal diseases has been increasing due to the aging population [2]. Approximately 266 million people worldwide are diagnosed with symptomatic spinal degenerative disease [3].

At present, the main treatment options for spinal diseases in the clinic are conservative and surgical treatment. For most patients with acute spinal injury, early surgical treatment is needed. Conservative treatment can be chosen for early-stage chronic degenerative spinal diseases; when the effect of conservative treatment is poor or cannot achieve the desired effect, surgical treatment can be chosen again [2]. Spinal fusion has become one of the common surgical methods for spinal diseases [4], because this method can effectively eliminate pain, relieve neurological symptoms, and stabilize the spine [5, 6]. Almost 500,000 patients undergo spinal fusions annually in the United States to treat degenerative disc disease and other spinal pathologies [7]. However, fusion failure is a common adverse outcome of surgery that can cause pain, neurological symptoms, spinal deformity and reduce internal fixation stability [4].

Previous studies have reported several factors that may affect spinal fusion, such as obesity (BMI $\geq$ 25 kg/m$^2$), smoking, graft type, vitamin D deficiency, surgical methods, and without the use of bone morphogenetic protein-2 (BMP-2). However, the results are still controversial. Niu et al.'s report suggests that patients who use BMP-2 have better fusion results than patients who do not use BMP-2 [8–13]. However, many other observational studies have not found a significant correlation between the use of BMP-2 and successful fusion [14–19]. Moreover, Zhang et al. reported that vitamin D deficiency could decrease spine fusion rates [20, 21], while Ravindra et al. found that there was no significant difference in spine fusion rates between vitamin D-deficient and non-vitamin D deficient patients [22].

To the best of our knowledge, there is no systematic review of all the risk factors that may affect spinal fusion. Therefore, we carried out a meta-analysis of risk factors reported in the literature. We also graded the evidence to better identify the risk factors affecting spinal fusion.

## Methods

### Standard protocol approvals, registrations, and patient consent

The review protocol was appropriately registered with PROSPERO (https://www.crd.york.ac.uk/prospero/) and reporting was conducted in strict accordance with guidelines from Cochrane Handbook, MOOSE (Meta-Analysis of Observational Studies in Epidemiology) [23], PRISMA (Preferred Reporting Items for Systematic Reviews and Meta-Analyses) [24]

and AMSTAR (Assessing the methodological quality of systematic reviews) Guidelines [25]. The MOOSE checklist is detailed in S1 Checklist.

## Search strategy

We conducted searches on three electronic databases (PubMed, EMBASE and Cochrane Library) for English articles published prior to January 6, 2023. These studies identified the risk factors affecting spinal fusion. In instances where multiple studies reported on the same cohort, priority was given to the most recently published study or the study encompassing the largest cohort size for inclusion in our analysis. We combined "spinal fusion", "fusion rate", and "risk factors" as keywords and searched PubMed and Cochrane Library using Medical Subject Terms (MESH), and Embase databases using Embase subject heading (Embase). The search terms included ("spinal surgery" or "spinal fusion" or "joint fusion") and ("fusion rate" or "fixation rate") and ("obesity" or "electric stimulation therapy" or "smoking" or "osteoporosis" or "vitamin D") (S1 Table).

Initial retrieval of citations was processed through Endnote X9, where duplicates were merged, identified, and subsequently removed through a manual process. The preliminary assessment of the literature involved an examination of titles and abstracts to screen for relevance to our study criteria. This was followed by a meticulous independent review of the full texts of preliminarily selected studies by the research team to confirm their suitability for our meta-analysis. This rigorous selection process culminated in the inclusion of 39 studies for comprehensive analysis.

## Selection criteria

Following the preliminary article screening, two investigators independently conducted a review and verification of the articles. Any disagreements were amicably resolved through discussion or by seeking the opinion of a third evaluator. Articles were considered eligible if they satisfied the following criteria based on population, intervention, comparison, outcome, and study design (PICOS) principles

1. Population: Patients with spinal diseases who have undergone spinal fusion surgery.

2. Intervention: Assess changeable patients and possible risk factors associated with surgery, including smoking, graft type, pedicle screw type, diabetes, vitamin D deficiency, number of fused levels, fusion column, and minimally invasive surgery (MIS), without the use of BMP-2.

3. Comparison: Analyzing the differences in modifiable risk elements among subjects with or without exposure.

4. Outcome: Identifying and quantifying related risk factors through the calculation of odds ratios (ORs) and their 95% confidence intervals (CIs).

5. Study design: Prospective or retrospective cohort study.

Exclusions were applied to literature reviews, animal experiments, non-English literature, and randomized controlled trials (RCTs). Furthermore, studies lacking sufficient data were also excluded.

## Data extraction and quality assessment

Two authors extracted data using a predesigned data extraction sheet. The specific extracted content was obtained by the relevant authors by reading the full text of the article and the

contents of the table. When the data was incomplete or missing, we tried to contact the corresponding author of the article to obtain the relevant data. Differences between researchers in the process of extracting data were resolved through discussion or negotiation with third parties. The following data were extracted: first author, publication year, country, type and site of surgery, observation period, sample size, study design, female proportion, mean age, measurements of fusion, mean follow-up period, significant variables.

Two authors evaluated each qualified study independently by the Newcastle–Ottawa Scale (NOS) [26], which encompasses 3 domains, including patient representation, exposure and outcome determination, and follow-up adequacy, with an overall score of 9 for each study. The NOS scores were then stratified into three qualitative tiers reflecting study quality: low (0–5 points), moderate (6–7 points), and high (8–9 points, indicative of a minimal bias risk) [27]. The quality evaluation results of the study included in this meta-analysis are shown in S2 Table.

## Evaluation of the strength of evidence

The grading of the strength of evidence in the identified associations for observational cohort studies was conducted utilizing a set of modified criteria [28]. When the P value of Egger's test was greater than 0.05, the total sample size was over 500, and interstudy heterogeneity $I^2$ was less than 50%, the association was deemed high-quality (class I) evidence. If two out of the three conditions were satisfied, the association was classified as class II (medium-quality) evidence. Meeting one of these three conditions resulted in a class III (medium-quality) evidence correlation. Failure to meet any of these three conditions indicated class IV (low-quality) evidence (S3 Table).

## Statistical analysis

All of our analyses were performed using Stata software (Stata version 16.0, College Station, Texas, USA). We analyzed the risk factors affecting spinal fusion, including patient-related factors (e.g., smoking, diabetes, and vitamin D deficiency). surgery-related risk factors (e.g., allograft, without the use of BMP-2, conventional pedicle screw (CPS) fixation, and posterolateral fusion, MIS, number of fused levels). The odds in each group were computed as $p/(1-p)$ where $p$ represents the proportion with exposure. The odds ratio (OR) was determined by dividing the odds in the fusion failure group by the odds in the comparator group. In the meta-analyses, study-specific log odds ratios were utilized as the outcome, and the aggregated estimates were then transformed into OR. If the OR > 1, it indicates a higher probability of fusion failure in the exposed group as opposed to the non-exposed group. Forest plots were utilized to present the ORs of individual studies as well as the pooled OR. The heterogeneity between studies was determined using the Cochrane Q test and $I^2$ test, with heterogeneity considered significant when $I^2 > 50\%$ [29]. Sensitivity analyses were conducted to evaluate the robustness of the findings by systematically excluding individual studies and subsequently pooling the estimates from the remaining studies through meta-analysis. Egger's test was used to evaluate publication bias for each risk factor by analyzing the relationship between the effect estimates and their variances. A P value of < 0.1 was deemed to signify a significant distinction [30]. All statistical tests were bidirectional, and $P < 0.05$ was considered statistically significant.

## Results

### Literature search and study characteristics

Out of 1,257 studies identified through a systematic literature search, 97 duplicate records were excluded, and 1,039 irrelevant studies were excluded after reviewing their titles and

abstracts. Next, we excluded 3 citations for which three could not be obtained in full-text form, and 118 studies were selected for review of the full paper. After a full-text review, we excluded 79 studies that did not have access to patient outcome data, non-population-based cohorts, meta-analyses, systematic reviews, RCTs, and non-English literature (S4 Table). Finally, this meta-analysis included 39 cohort studies [8–22, 31–54], comprising 7,145 participants satisfied the inclusion criteria (Fig 1).

Table 1 displays the baseline characteristics of the studies included in the analysis. All studies were published between 1996 and 2022, and 24 (60%) of the studies were published in 2013 or later [9, 12–15, 17, 19–22, 37–41, 43, 45, 46, 48–51, 53, 54]. The studies involved 11 countries with an average sample size of 183, and the average follow-up time was 31 months (range 6–183 months). Out of the studies analyzed, 30 (76.9%) studies achieved an NOS score of $\geq 8$ (S2 Table) [9–12, 14–16, 20–22, 32–38, 40–42, 44–49, 51–54]. In 25 (64%) [10–13, 15–21, 34–37, 39–41, 43, 46, 48, 49, 51, 53, 54] studies, computed tomography (CT) scans were used to assess fusion. The standards for spinal fusion were defined in 27 (69%) studies [10, 15–22, 31, 32, 34, 37–39, 41–44, 46–50, 52–54].

Fusion rates of spinal surgery ranged from 65% to 100%, and the combined random-effect model fusion rate was 89.2% (95% CI, 87.4% to 91.1%; $I^2 = 86.9\%$, P < 0.001) (S1 Fig). Therefore, to explore the source of between-study heterogeneity, we stratified by some baseline study-level factors (all P < 0.001). Among these, we found fusion rates that were significantly different; for example, the combined fusion rate of studies with a female proportion below 50% was found to be 85.1% (95% CI, 79.4% to 90.8%; $I^2 = 91.7\%$, P < 0.001), which was significantly less than that in other studies. In addition, in the stratified analysis of surgical sites and surgical methods, we found that the fusion rate of cervical surgery was 92% (95% CI, 89.7% to 94.3%; $I^2 = 73.7\%$, P < 0.001), which was Higher than the fusion rate of lumbar surgery. And

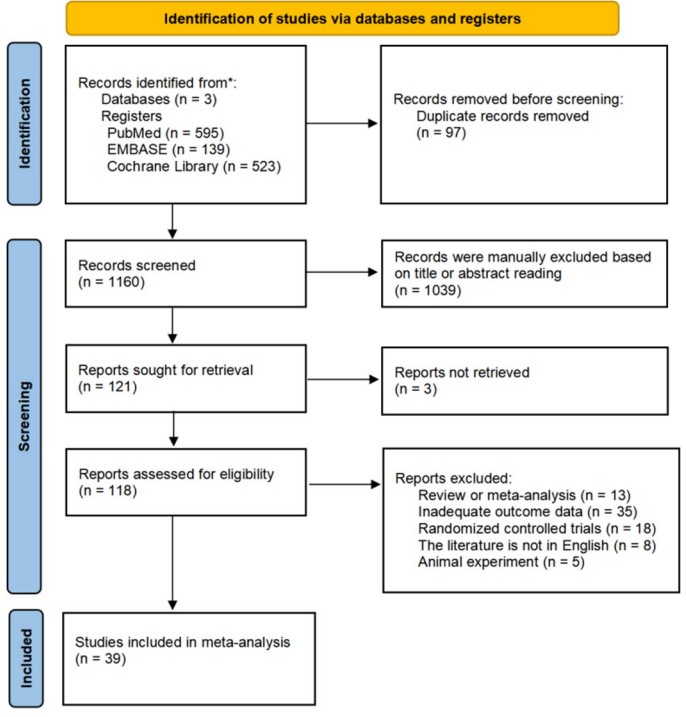

**Fig 1. Flowchart of the study selection.**

**Table 1. Characteristics of studies included in meta-analysis.**

| First author | Year | Country | Observation Period | Study Design | Sample size | Female % | Mean age | Measurements of fusion | Mean Follow-up period |
|---|---|---|---|---|---|---|---|---|---|
| Glassman, S. D. | 2000 | USA | 1992–1996 | Retrospective | 357 | 43.0 | 43.9 | 3D-CT | 24 months |
| Bose, B. | 2001 | USA | NR | Retrospective | 106 | 55.7 | 50.1 | X-ray | ≥12 months |
| Samartzis, D. | 2003 | USA | NR | Retrospective | 80 | 36.3 | 49.0 | Radio graphs | 16 months |
| Glassman, S. D. | 2003 | USA | NR | Retrospective | 137 | 60.5 | 60.3 | CT | 12 months |
| Gerszten, P. C. | 2011 | USA | 2005–2007 | Retrospective | 99 | 42.8 | 42.8 | MRI and CT | 24 months |
| Hoffmann, M. F. | 2012 | USA | 2003–2009 | Retrospective | 1398 | 58.9 | 60.0 | CT and X-ray and MRI | 183 months |
| Luszczyk, M. | 2013 | USA | NR | Retrospective | 573 | NR | NR | Radio graphs | 24 months |
| Urrutia, J. | 2013 | Chile | 2004–2010 | Retrospective | 47 | 76.6 | 46.5 | CT | 50.1 months |
| Frenkel, M. B. | 2013 | USA | 1997–2012 | Retrospective | 45 | 57.7 | 50.0 | X-ray and CT | 161 months |
| Adams, C. L. | 2014 | Australia | 2007–2010 | Retrospective | 70 | 50.0 | 55.4 | Radio graphical | 12 months |
| Tan, B. | 2015 | China | 2007–2010 | Retrospective | 146 | 41.1 | 64.7 | Radio graphs | 24 months |
| Yang, Y. | 2016 | China | 2011–2014 | Retrospective | 132 | 32.6 | 44.2 | CT | 24 months |
| Zhang, Y. H. | 2017 | China | 2007–2015 | Retrospective | 32 | NR | 6.80 | CT | 45 months |
| Phan, K. | 2018 | Australia | NR | Retrospective | 137 | 52.6 | 56.7 | CT | 12 months |
| Weng, F. | 2018 | China | 2015–2017 | Retrospective | 80 | 71.3 | 65.6 | NR | 6 months |
| Nourian, A. A. | 2019 | USA | NR | Retrospective | 93 | 67.0 | 65.0 | CT | >12 months |
| Niu, S. | 2020 | NR | 2007–2017 | Retrospective | 927 | 59.0 | 63.8 | CT | 6 months |
| Wu, F. L. | 2020 | China | NR | Retrospective | 50 | 48.0 | 61.2 | MRI and CT | 30 months |
| Son, H. J. | 2021 | USA | 2009–2019 | Retrospective | 121 | 76.0 | 68.2 | CT | >12 months |
| Tan, Y. | 2021 | Japan | 2017–2020 | Retrospective | 98 | 64.3 | 69.5 | CT | 33.5 months |
| Tannoury, C. | 2021 | USA | 2008–2019 | Retrospective | 220 | 62.8 | 66.0 | CT | 12 months |
| Wang, H. | 2021 | China | NR | Retrospective | 153 | 66.8 | 50.1 | CT | 40.8 months |
| Li, Z. | 2022 | China | 2017–2019 | Retrospective | 77 | 52.0 | 44.6 | CT | 12 months |
| Bishop, R. C. | 1996 | USA | 1991–1995 | Prospective | 132 | 54.6 | 45.2 | X-ray | 31 months |
| Tuli, S. K. | 2004 | USA | 1995–1999 | Prospective | 57 | 47.4 | 52.2 | Radio graphs | 6 months |
| Suchomel, P. | 2004 | Czech | 1998–2000 | Prospective | 79 | 37.9 | 47.8 | X-rays | 48 months |
| Burkus, J. K. | 2004 | USA | NR | Prospective | 46 | NR | NR | Radio assessments | 24 months |
| Cammisa Jr, F. P. | 2004 | USA | NR | Prospective | 120 | 49.0 | 48.0 | X-ray | 24.5 months |
| Burkus, J. K. | 2005 | USA | 1998–2001 | Prospective | 131 | 61.1 | 41.5 | Radio graphs and CT | ≥24 months |
| Joseph, V. | 2007 | Canada | 2003–2005 | Prospective | 33 | 39.4 | 49.7 | CT | 25 months |
| Frantzén, J. | 2011 | Finland | 1996–1998 | Prospective | 17 | 70.5 | 49.4 | CT and X- ray and MRI | 132 months |
| Wu, Z. X. | 2012 | China | 2004–2009 | Prospective | 157 | 61.2 | 62.1 | Radio graphs | 43 months |
| Ravindra, V. M. | 2015 | USA | 2011–2012 | Prospective | 133 | 44.0 | 57.0 | CT | >12 months |
| Burkus, J. K. | 2017 | USA | NR | Prospective | 710 | NR | NR | X-ray | 24 months |
| Moazzeni, K. | 2018 | Iran | 2014–2015 | Prospective | 96 | 62.5 | 57.8 | CT and X-ray | 12 months |
| Ravindra, V. M. | 2019 | USA | 2011–2012 | Prospective | 58 | 41.4 | 57.1 | X-ray | ≥12 months |
| Srour, R. | 2020 | France | 2017–2018 | Prospective | 53 | 50.9 | 65.0 | CT | 12 months |
| Hyun, S. J. | 2021 | Korea | NR | Prospective | 76 | 44.7 | 63.4 | X-ray and CT | 12 months |
| Zhang, W. | 2022 | China | 2018–2020 | Prospective | 69 | 61.5 | 54.6 | CT | 6 months |

| First author | Year | Fusion definition | significant variables | Fusion rate (%) | Surgical sites | Surgical types | |
|---|---|---|---|---|---|---|---|
| Glassman, S. D. | 2000 | NR | Smoking | Smokers: 79.3%, Nonsmokers: 85.8% | Lumbar | Posterior fusion | |

*(Continued)*

**Table 1.** (Continued)

| | | | | | | |
|---|---|---|---|---|---|---|
| Bose, B. | 2001 | Trabecular bony bridging across the disk space and lack of motion on flexion extension. | Smoking | Nonsmoking: 96.67%, smoking: 97.83% | Cervical | ACDF |
| Samartzis, D. | 2003 | A bony bridge incorporated the graft and the adjacent end plates and no radiolucencies or motion. | Autograft, allograft | Allograft: 94.3%; autograft: 100% | Cervical | ACDF |
| Glassman, S. D. | 2003 | NR | IDDM, NIDDM | NIDDM: 78.26%/ IDDM: 74.28%, Control: 94.59% | Lumbar | PLIF |
| Gerszten, P. C. | 2011 | Bridging bone that traversed from end plate to end plate, with no motion detected on flexion extension lateral radio graphs. | rhBMP-2 | With rhBMP-2: 95.5%, Without rhBMP-2: 92.5% | Lumbar | Interbody fusions in the lumbosacral spine |
| Hoffmann, M. F. | 2012 | NR | DBM with rhBMP-2 | rhBMP-2: 95.7%/ DBM: 86.9, Autograft: 84.8% | Lumbar | PLIF, TLIF |
| Luszczyk, M. | 2013 | Bony trabecular bridging between the graft and vertebral body, and motion was absent. | Smoking | Smoking: 91%, control: 91.6% | Cervical | ACDF |
| Urrutia, J. | 2013 | Trabeculae crossing the graft-vertebral body interface on both sides of the graft either. | Smoke | Smoke: 91.7% | Lumbar | Circumferential lumbar spinal fusion |
| Frenkel, M. B. | 2013 | The presence of motion on flexion-extension radio graphs. | rhBMP-2 | With rhBMP-2: 100%, Without rhBMP-2: 83% | Cervical | Anterior cervical fusion |
| Adams, C. L. | 2014 | NR | rhBMP-2, LBC | rhBMP-2: 94.1%, LBC: 89.5% | Lumbar | PLIF or TLIF |
| Tan, B. | 2015 | NR | rhBMP-2, ICBG | rhBMP-2: 87.7%, ICBG: 74% | Cervical | ACDF |
| Yang, Y. | 2016 | Evidence of continuous bridging bone between the adjacent end plates of the involved motion segment, radiolucent lines at 50% or less of the graft vertebra interfaces. | Two-level or single level ACDF | Single level: 94.6%, Two-level: 92.7% | Cervical | ACDF |
| Zhang, Y. H. | 2017 | The lack of hardware failure and presence of continuous bridging trabecular bone between the dorsal elements of the C1 and C2 on CT scans. | Structural allograft or autograft | Allograft: 94%, autograft: 100% | Cervical | Atlantoaxial fusion |
| Phan, K. | 2018 | Bridging trabecular formation across the intervertebral disk space with the absence of radiolucency spanning more than half of the implant. | Smoking | Smoking: 69.6%, no smoking: 85.1% | Lumbar | ALIF |

(*Continued*)

**Table 1.** (Continued)

| | | | | | | |
|---|---|---|---|---|---|---|
| Weng, F. | 2018 | Complete fusion and remodeling after intervention with newly-formed trabeculae; complete bone block after intervention. | EPS, CPS | EPS: 90%, CPS: 50% | Lumbar | Lumbar short-segment fixation and fusion |
| Nourian, A. A. | 2019 | NR | LLIF with rhBMP-2 | 1-level LLIF with rhBMP-2: 92%, 2-level LLIF with rhBMP-2: 86% | Lumbar | LLIF |
| Niu, S. | 2020 | NR | RhBMP-2 | RhBMP-2: 92.5%, no rhBMP-2: 71.4% | Lumbar | TLIF |
| Wu, F. L. | 2020 | NR | MIDLF, MI-TLIF | MIDLF: 94%, MI-TLIF: 88% | Lumbar | MIDLF, MI-TLIF |
| Son, H. J. | 2021 | Lenke grade B and BSF grade-3. | ICBG, E.BMP-2 | ICBG: 97.1%, E. BMP-2: 100% | Lumbar | LIF and PLF |
| Tan, Y. | 2021 | NR | Simultaneous single-position O-arm-navigated OLIF and PPS, MI-PLIF/TLIF | OLIF and PPS: 96.8%, MI-PLIF/TLIF: 94.2% | Lumbar | MIS-ATP-lumbar fusions |
| Tannoury, C. | 2021 | Examining consecutive sagittal and coronal cuts for continuous bony bridges. | Smoke, L5-S1 | Smoke: 95.3%, L5-S1: 95.2% | Lumbar | OLIF and PPS, MI-PLIF/TLIF |
| Wang, H. | 2021 | Mental ROM less than 3° in X-ray and continuous bone bridge demonstrated in CT imaging. | Male, smoke | Smoke: 87.84% | Cervical | CDR and ACDF |
| Li, Z. | 2022 | Unilateral or bilateral grade I or II fusion. | Modified facet joint fusion or posterolateral fusion | MFF: 94.3%, PLF: 76.2% | Lumbar | Modified facet joint fusion or posterolateral fusion |
| Bishop, R. C. | 1996 | Bony trabeculae were seen crossing the involved interspace. | Autograft tricortical iliac and the allograft tricortical iliac | Single-level ACDF: (autograft: 97%, allograft: 87%), multiple level interbody fusion: (autograft: 100%, allograft: 89%) | Cervical | Single-level or multiple-Level ACDF |
| Tuli, S. K. | 2004 | Presence of any trabeculae bridging between the vertebral body and allograft at the upper and lower aspects. | One level or two-level corpectomy | Cephalad aspect of the graft-host interface: 92%, caudad aspect: 93%, one-level corpectomy: 86%, two-level corpectomy: 100% | Cervical | Cervical decompressive corpectomy and reconstruction |
| Suchomel, P. | 2004 | Complete bridging of trabeculae between adjacent vertebral bodies and bone graft. | Number of fused levels, smoke, autologous, allogenic bone grafts | Autografts: 94.6%, allografts: 85.5%, smoke: 92.4% | Cervical | One- or two-level ACDF |
| Burkus, J. K. | 2004 | NR | rhBMP-2, ICBG | rhBMP-2-treated: 100%, auto graft-treated: 68.4% | Lumbar | ALIF |
| Cammisa Jr, F. P. | 2004 | NR | Grafton®, auto graft | Grafton® side: 52%, auto graft side: 54% | Lumbar | Posterolateral spine fusion |

(*Continued*)

**Table 1.** (Continued)

| Burkus, J. K. | 2005 | (1) The presence of bridging trabecular bone connecting vertebral bodies through or around dowels, (2) no radiolucent area involving > 50% of the interface between dowels and end plates. | rhBMP-2/ACS, ICBG | rhBMP-2-treated: 98.5%, autograft-treated: 76.1% | Lumbar | Single-level ALIF |
|---|---|---|---|---|---|---|
| Joseph, V. | 2007 | The presence of bridging bone through the cage or external to it. | rhBMP-2 | With BMP: 94.4%, Without BMP: 89.4% | Lumbar | PLIF and TLIF |
| Frantzén, J. | 2011 | Bridging of bone between the transverse processes in addition to the incorporation of bone between the transverse processes. | BAG | BAG: 90%, Autologous Bone: 100% | Lumbar | Posterolateral spondylodesis |
| Wu, Z. X. | 2012 | Clear trabecular bone bridging across the segment to be fused, translation of 3 mm or less and angulation of < 5° on flexion-extension radio graphs. | EPS, CPS | EPS: 92.5%, CPS: 80.5% | Lumbar | Transpedicle fixation |
| Ravindra, V. M. | 2015 | The presence of bone trabeculation, without evidence of instrumentation loosening or breakage, and no observed motion between the graft and instrumentation. | Vitamin D | 84% | spine | Spinal fusion |
| Burkus, J. K. | 2017 | NR | rhBMP-2, ICBG | rhBMP-2-treated: 99.4%, autograft-treated: 87.2% | Cervical | Single-level anterior cervical arthrodesis |
| Moazzeni, K. | 2018 | Bridging bone remodeling across the transverse processes between the adjacent vertebrae. | Diabetic and non-diabetic patients after lumbar fusion | DM: 53%, control: 78% | Lumbar | Bilateral facet fusion |
| Ravindra, V. M. | 2019 | The presence of trabeculated bone, without evidence of hardware loosening or failure, and no observed motion between vertebral segments on X-rays. | Low vitamin D | Normal vitamin D: 76.47%, low vitamin D: 75.60% | Cervical | Anterior, posterior, or combined spinal fusion |
| Srour, R. | 2020 | Any sign of bony fusion inside or posterior to the device when viewing the postoperative CT scan. | Facet arthrodesis with or without PLIF | Facet arthrodesis with PLIF: 75.7%, non-PLIF: 88.7% | Lumbar | Facet osteosynthesis |
| Hyun, S. J. | 2021 | Less than 5 degrees of angular motion on flexion and extension radio graphs. | rhBMP-2 | DBM with rhBMP-2: 82.85%, DBM: 78.12% | Lumbar | TLIF |

(*Continued*)

**Table 1.** (Continued)

| Zhang, W. | 2022 | Bridging bone bonding with both adjacent vertebral bodies. | Vitamin K2 + Vitamin D3 | Vitamin K2 + Vitamin D3: 91.18%, control: 71.43% | Lumbar | TLIF or PLIF |
| --- | --- | --- | --- | --- | --- | --- |

Abbreviations: ACDF = Anterior cervical discectomy and fusion; ACS = absorbable collagen sponge; ALIF = Anterior lumbar interbody fusion; BAG = bioactive glass; BMI = body mass index; BSF = Brantigan, Steffee and Fraser; CDR = Cervical disc replacement; CPS = conventional pedicle screws; CT = computed tomography; DBM = demineralized bone matrix; DM = diabetes mellitus; E.BMP-2 = E.coli-derived rhBMP-2; EPS = expandable pedicle screws; HA = hydroxyapatite; HO = heterotopic ossification; ICAG = iliac crest autograft; ICBG = iliac crest bone graft; IDDM = insulin-dependent diabetes mellitus; JOA = Japanese Orthopedic Association; LBC = local bone graft; LBP = low back pain; LLIF = Lateral lumbar interbody fusion; MFF = modified facet joint fusion; MIDLF = midline lumbar fusion; MI-PLIF/TLIF = Minimally invasive posterior or transforaminal lumbar interbody fusion; MIS-ATP = minimally invasive antepsoas; MI-TLIF = minimally invasive transforaminal lumbar interbody fusion; MRI = magnetic resonance imaging; NDI = Neck Disability Index; NIDDM = non-insulin dependent diabetes mellitus; NR = not reported; OLIF and PPS = Oblique lateral interbody fusion and percutaneous pedicle screw; OP-1 = Osteogenic Protein-1; PLF = posterolateral fusion; PLIF = posterior lumbar interbody fusion; RCT = Randomized Controlled Trial; rhBMP-2 = recombinant human bone morphogenetic protein-2; ROM = range of motion; RTC = rectangular titanium cage; TLIF = transforaminal lumbar interbody fusion; VAS = visual analog scale; W-TLIF = TLIF through Wiltse approach.

in lumbar fusion surgery, the fusion rate of lateral approach was significantly higher than that of other approaches (Rate, 95.3%; 95% CI, 90.8% to 99.7%; $I^2$ = 70.2%, P = 0.035) (S5 Table).

## Risk factors and strength of evidence

Our study included the effects of patient-related and surgery-related risk factors (Fig 2) on spinal fusion. High-quality (Class I) evidence showed that patients with a smoking habit and without the use of BMP-2 were at higher risk for fusion failure. Medium-quality (Class II or

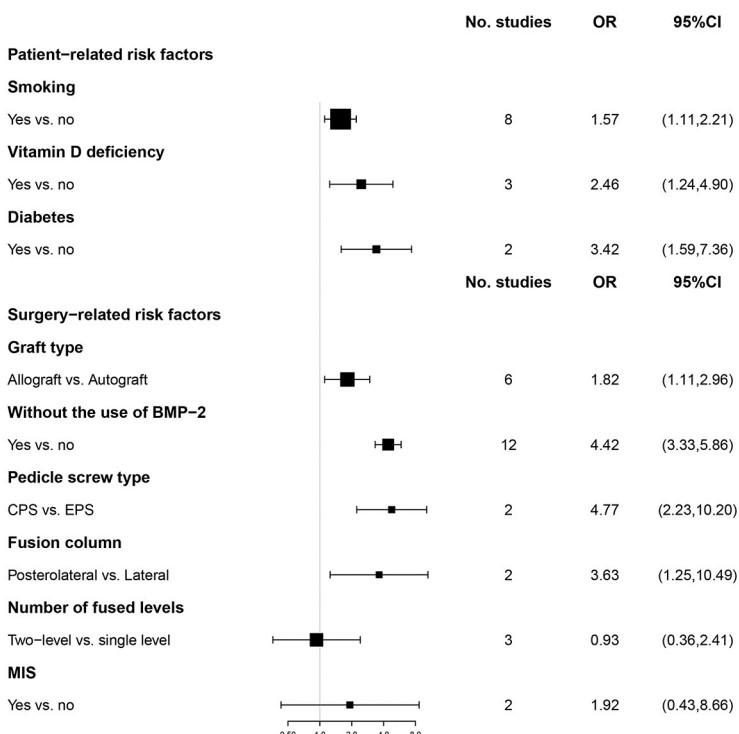

**Fig 2. Meta-analyses of the association between patient-related risk factors and surgery-related risk factors.**

**Table 2. Significant and non-significant risk factors associated with spinal fusion failure.**

| Significant factors | No. of Studies | No. of Patients | OR (95% CI) | I$^2$, % | P value | Egger's test P value |
|---|---|---|---|---|---|---|
| Smoking | | | | | | |
| No | | | Ref. | | | |
| Yes | 8 | 1672 | 1.57 (1.11 to 2.21) | 0.0 | 0.010 | 0.702 |
| Graft type | | | | | | |
| Autograft | | | Ref. | | | |
| Allograft | 6 | 460 | 1.82 (1.11 to 2.96) | 25.7 | 0.018 | 0.024 |
| Without the use of BMP-2 | | | | | | |
| No | | | Ref. | | | |
| Yes | 12 | 3802 | 4.42 (3.33 to 5.86) | 36.2 | 0.000 | 0.593 |
| Vitamin D deficiency | | | | | | |
| No | | | Ref. | | | |
| Yes | 3 | 260 | 2.46 (1.24 to 4.90) | 13.5 | 0.010 | 0.822 |
| Pedicle screw type | | | | | | |
| EPS | | | Ref. | | | |
| CPS | 2 | 237 | 4.77 (2.23 to 10.20) | 47.5 | 0.000 | / |
| Diabetes | | | | | | |
| No | | | Ref. | | | |
| Yes | 2 | 233 | 3.42 (1.59 to 7.36) | 0.0 | 0.002 | / |
| Fusion column | | | | | | |
| Lateral | | | Ref. | | | |
| Posterolateral | 2 | 130 | 3.63 (1.25 to 10.49) | 0.0 | 0.017 | / |
| **Non-significant factors** | **No. of Studies** | **No. of Patients** | **OR (95% CI)** | **I$^2$, %** | **P value** | **Egger's test P value** |
| Number of fused levels | | | | | | |
| Single | | | Ref. | | | |
| Two | 3 | 282 | 0.93 (0.36 to 2.41) | 19.6 | 0.887 | 0.025 |
| MIS | | | | | | |
| No | | | Ref. | | | |
| Yes | 2 | 148 | 1.92 (0.43 to 8.66) | 0.0 | 0.396 | / |

Abbreviations: BMP-2, bone morphogenetic protein-2; CI, confidence interval; CPS, conventional pedicle screws; EPS, expandable pedicle screws; MIS, minimally invasive surgery; OR, odds ratio; Ref, Reference group;

III) evidence showed that fusion failure was significantly associated with vitamin D deficiency, diabetes, allograft, CPS fixation, and posterolateral fusion. Additionally, moderate-quality (Class II) evidence revealed nonsignificant correlations between MIS or the number of fused levels (two-level versus single-level) and fusion failure (Table 2 and S6 Table).

### Patient-related risk factors

**Smoking.** This meta-analysis showed that patients who smoked were at higher risk for fusion failure. The combined OR of 8 studies [32, 36, 38, 41, 44, 46, 48, 49] was 1.57 (95% CI, 1.11 to 2.21; I$^2$ = 0.0%) (Fig 3). The trim-and-fill method was used to assess the robustness of the results, and we did not find potentially missing studies (S6 Table).

**Diabetes.** We included 2 studies [35, 39] that evaluated the effect of diabetes on spinal fusion, and the combined OR was 3.42 (95% CI, 1.59 to 7.36; I$^2$ = 0.0%) (Fig 4). We used the trim-and-fill method to adjust the publication bias and found that there was only one missing

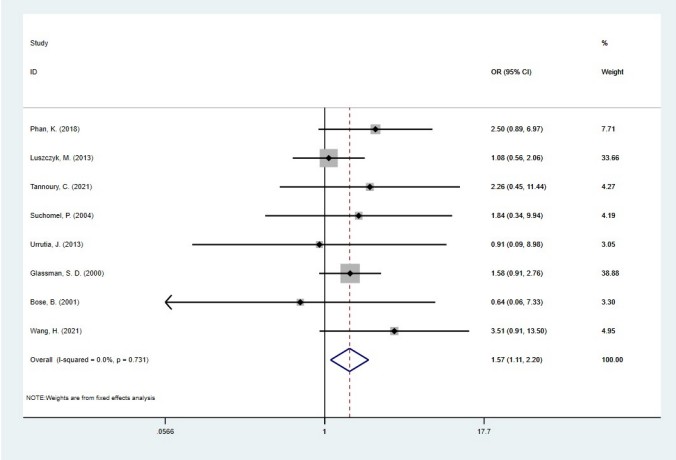

**Fig 3. Odds ratio (OR) for association between smoking and fusion rate.**

potential study in the funnel plots. The OR corrected for publication bias was 2.71 (95% CI, 1.36 to 5.41), which was largely consistent with our results (S6 Table).

**Vitamin D deficiency.** Three studies [20–22] reported the effect of vitamin D deficiency on spinal fusion. In our outcome, the risk of fusion failure in patients with vitamin D deficiency was significantly higher than that in patients without vitamin D deficiency (OR, 2.46; 95% CI, 1.24 to 4.90) (Fig 5), and we did not observe significant heterogeneity ($I^2$ = 13.5%).

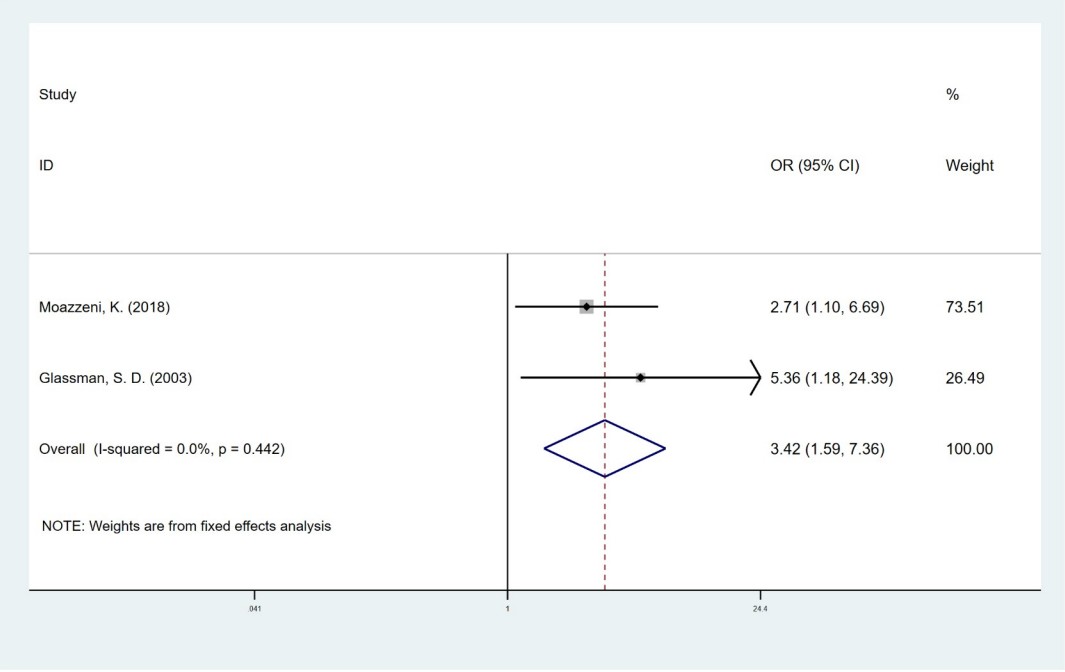

**Fig 4. Odds ratio (OR) for association between diabetes and fusion rate.**

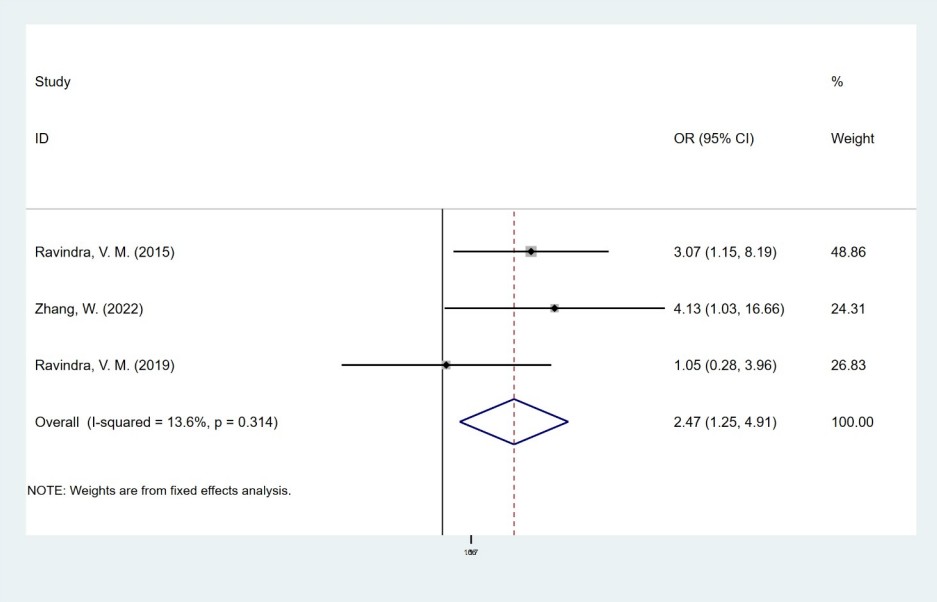

**Fig 5. Odds ratio (OR) for association between vitamin D deficiency and fusion rate.**

We used the trim-and-fill method to adjust the publication bias and did not find missing potential studies (S6 Table).

## Surgery-related risk factors

**BMP-2.** We included 12 studies [8–12, 14–19, 45] that evaluated the effect of BMP-2 on spinal fusion (Fig 6). We found high-quality (Class I) evidence for a significant association between fusion failure and without the use of BMP-2 versus the use of BMP-2 (OR, 4.41; 95% CI, 3.33 to 5.86; $I^2$ = 36.2%) (Table 2 and S6 Table). From our analysis, we found that the

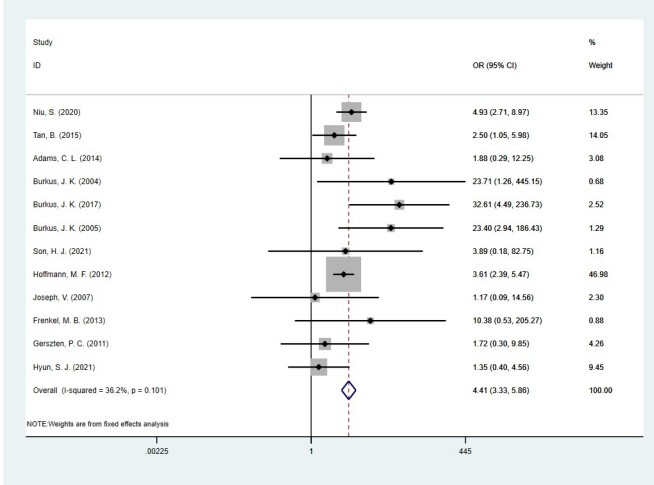

**Fig 6. Odds ratio (OR) for association between without the use of BMP-2 (yes vs. no) and fusion rate.**

pooled OR was not significantly affected after removing any single study (lowest OR, 3.46; 95% CI, 2.49 to 4.79, highest OR, 3.88; 95% CI, 2.61 to 5.76) (S7 Table). The trim-and-fill method was used to adjust the publication bias, revealing a solitary absent potential study in the funnel plots. The OR corrected for publication bias was 3.58 (95% CI, 2.70 to 4.75), which was basically consistent with our results (S6 Table).

**Graft type.** The combined results of 6 studies [31, 33, 34, 42, 44, 53] showed that compared with autografts, the risk of fusion failure of allografts was higher (OR, 1.82; 95% CI, 1.11 to 2.97; $I^2$ = 25.7%) (Fig 7). However, after adjusting for publication bias by the trim-and-fill method, the pooled OR was 1.28 (95% CI, 0.79 to 2.07) (S6 Table), which was different from our results.

**Pedicle screw type.** Data from 2 studies [50, 52] suggested that there was a higher risk of fusion failure with CPS fixation than with expandable pedicle screw (EPS) fixation (OR, 4.77; 95% CI, 2.23 to 10.20; $I^2$ = 47.5%) (Fig 8). We used the trim-and-fill method to adjust the publication bias, and the OR corrected for publication bias was 2.98 (95% CI, 1.56 to 5.81), which was essentially in line with our results (S6 Table).

**Fusion column.** The combined results of 2 studies [37, 43] suggested that compared with posterolateral fusion, lateral fusion may increase the risk of fusion failure (OR, 3.63; 95% CI, 1.25 to 10.49; $I^2$ = 0.0%) (Fig 9). The trim-and-fill method was employed to evaluate the robustness of the outcome, and only one potential study was identified. The OR corrected for publication bias was 2.40 (95% CI, 0.98 to 5.88), which diverges from the outcomes obtained in our study (S6 Table).

**Number of fused levels.** We included 3 studies [40, 47, 54] that evaluated the effect of the number of fused levels on spinal fusion. Their results showed that there was no significant association between fusion failure and two-level fusions versus single-level fusion (OR, 0.93; 95% CI, 0.36 to 2.41; $I^2$ = 19.6%) (Fig 10). We used the trim-and-fill method to adjust the publication bias and did not find missing potential studies (S6 Table).

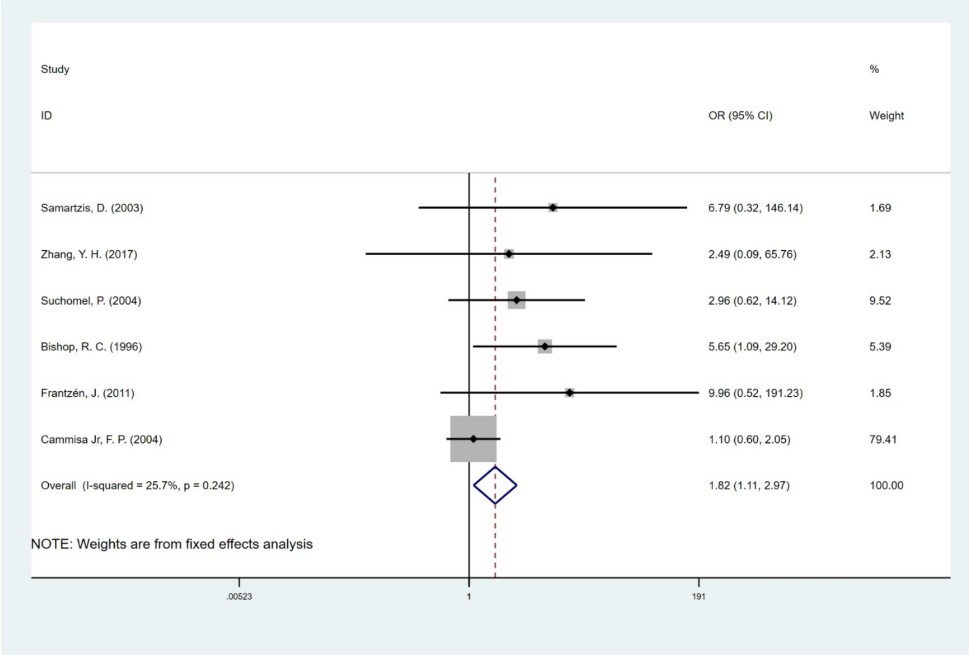

**Fig 7. Odds ratio (OR) for association between graft type (allograft vs. autograft) and fusion rate.**

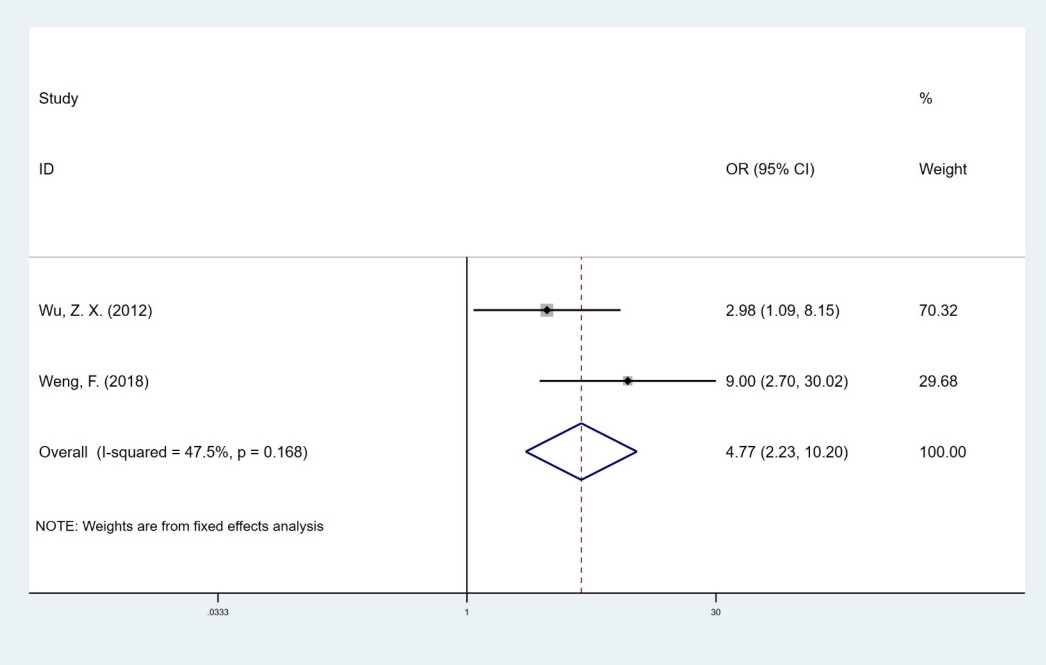

**Fig 8. Odds ratio (OR) for association between pedicle screw type (CPS vs. EPS) and fusion rate.**

**MIS.**  Two studies [13, 51] reported the effect of MIS on spinal fusion; however, no significant correlation was observed between the two (OR, 1.92; 95% CI, 0.43 to 8.66; $I^2$ = 0.0%) (Fig 11). The trim-and-fill method was employed to address publication bias, resulting in the identification of only one potential study that was missing. The OR corrected for publication

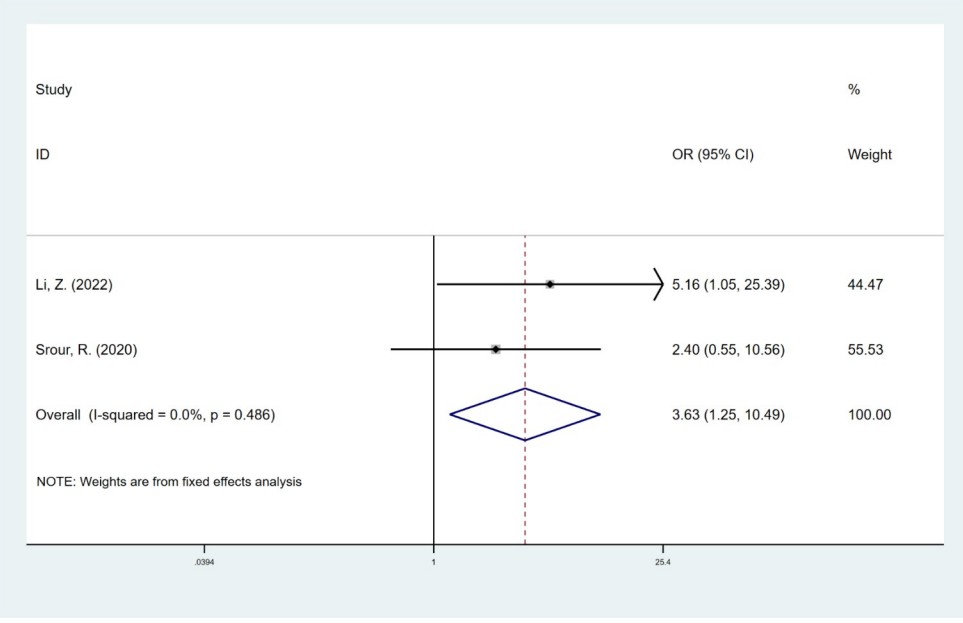

**Fig 9. Odds ratio (OR) for association between fusion column (posterolateral vs. lateral) and fusion rate.**

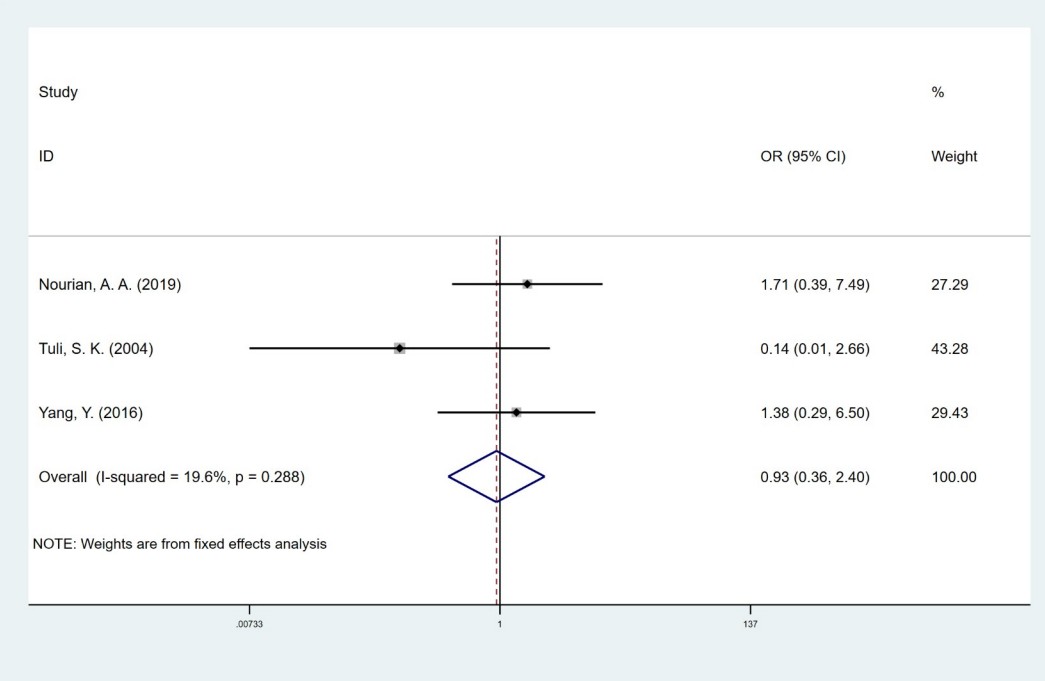

**Fig 10. Odds ratio (OR) for association between number of fused levels (two vs. single) and fusion rate.**

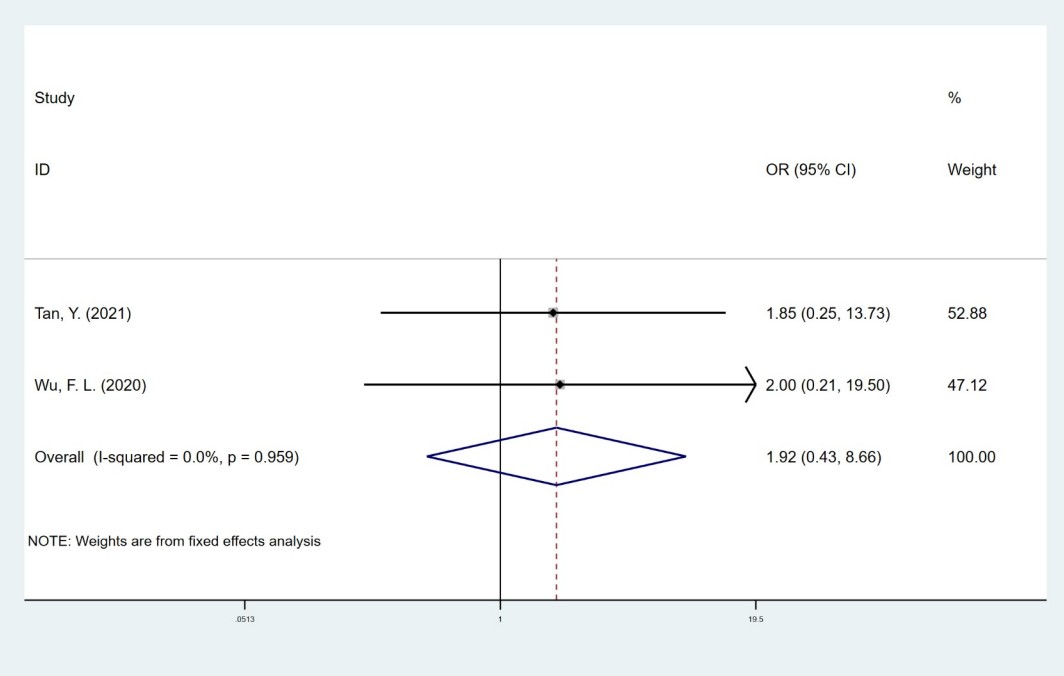

**Fig 11. Odds ratio (OR) for association between MIS and fusion rate.**

bias by the trim-and-fill method was 1.85 (95% CI, 0.53 to 6.46), which was basically consistent with our results (S6 Table).

### Sensitivity analyses and publication bias

We used leave-one-out sensitivity analysis to evaluate the stability of the results for factors reported by more than two articles. The results showed that the pooled ORs all remained similar across these analyses for both patient-related and surgery-related risk factors (S7 Table). Furthermore, funnel plots were employed to evaluate the potential presence of publication bias associated with these risk factors (S2–S10 Figs) and we didn't find any obvious bias.

## Discussion

### Principal findings

This meta-analysis was designed to identify risk factors affecting spinal fusion and to grade the level of evidence, and a total of 39 studies were included. We identified 3 patient-related risk factors, including smoking, diabetes, and vitamin D deficiency, and four surgery-related risk factors, including allografting, without the use of BMP-2, CPS fixation, and posterolateral fusion.

The meta-analysis revealed that MIS or the number of fused levels (two-level versus single-level) was not significantly linked to fusion failure. However, we cannot dismiss these factors as potential risk factors, as some studies have demonstrated a significant association with a low fusion rate [13, 47, 51]. Hence, it is advisable to carry out additional clinical studies on these variables.

### Potential mechanisms

The underlying mechanisms of various factors affecting spinal fusion have not been clarified until now. Smoking has been shown to impair skeletal healing and metabolism. Experiments have shown that nicotine can reduce neovascularization and inhibit osteoblast differentiation, resulting in bone healing defects [55–57]. In our study, diabetes was one of the significant risk factors for spinal fusion (OR, 3.42; 95% CI, 1.59 to 7.36). Diabetes is a multiorgan disease, and its complications may lead to multisystem organ failure, resulting in poor surgical outcomes [58]. Additionally, studies have confirmed that vitamin D levels are significantly associated with bone mineral density; thus, vitamin D deficiency may result in bone nonunion or prolonged fusion time [20, 21]. This is consistent with our findings (OR, 2.47; 95% CI, 1.25 to 4.91).

BMP-2 is an osteoinductive growth factor that belongs to the transforming growth factor-β (TGF-β) superfamily; it can stimulate pluripotent cells to form bone, and it is the only bone inducer with level I clinical evidence [8, 11, 59]. BMP-2 was introduced in the medical scenario to promote bone healing with the proposal of less morbidity compared to the usual methods of bone graft harvest [59]. Niu et al. reported that patients who used BMP-2 had better fusion rates than patients who were without the use of BMP-2 [8–13]. Moreover, we found that even if there are some complications, autografts remain the gold standard for interbody grafts in spinal fusion [44]. It has been demonstrated that autografts contain viable osteoblasts and osteogenic precursor cells that can contribute to the formation of new bone, thus improving the fusion rate [60]. However, allografts are considered to have high osteoconductive properties [61], weak osteoinductive potential, and non-osteogenic properties [62, 63]. Therefore, autografts provide better conditions for bone fusion and a higher fusion rate than allografts, which was also fully reflected in our research.

Additionally, Wu et al. and Weng et al. showed that compared with EPS fixation, CPS fixation has lower stability than internal fixation [50, 52]. EPS can fix the vertical axial section through the front expansive effect [64], thus forming triangular support [65] and significantly enhancing screw bonding [66]; in parallel, the surrounding bone trabecula is appropriately compressed, which consequently enhances both bone density and the stability of internal fixation [67]. Hence, the fusion rate of EPS fixation surpasses that of CPS fixation.

### Implications

Our study comprehensively shows the risk factors that may affect spinal fusion, and the identification of these factors can help clinicians to conduct a more comprehensive preoperative risk assessment of patients and early intervention, while developing appropriate surgical strategies for patients to reduce the risk of fusion failure. Therefore, conducting extensive prospective cohort studies is essential to validate these findings.

### Strengths

The advantages of our study are as follows. First, to the best of our knowledge, this is the first meta-analysis to assess all the risk factors that may affect spinal fusion. It provides the latest and most comprehensive evidence of risk factors affecting spinal fusion, including smoking, diabetes, vitamin D deficiency, allograft, without the use of BMP-2, CPS fixation, and posterolateral fusion. Second, to maximize the retrieval of original literature meeting the inclusion criteria and mitigate publication bias in the combined results, we developed an extensive database search strategy encompassing PubMed, Cochrane Library, and Embase, without imposing any date restrictions. Third, we also calculated the pooled fusion rate of spinal surgery by a random effect model and analyzed some factors at the baseline study level. Fourth, we assessed the strength of correlation for each risk factor (from Class I to Class IV) by considering factors such as sample size, Egger's P value, and heterogeneity. Lastly, we employed a range of rigorous methods to assess the robustness of our findings, such as sensitivity analysis and the trim-and-fill method.

### Limitations

The study has the following limitations despite its abovementioned strengths. First, our data sources were based on cohort studies, and the related risk factors that lead to an increase in the risk of fusion failure are diverse and complex; they were to some extent subject to selection bias. Second, few studies were involved in the analysis of some of the risk factors, making it difficult to accurately assess their relationship with spinal fusion, highlighting the need for future high-quality large cohort studies. Finally, given the absence of established gold standards or guidelines for quantitatively evaluating the strength of risk factor meta-analysis evidence, we employed three criteria (Egger's P value, sample size, and $I^2$ statistics) to classify the level of evidence intensity in accordance with existing literature.

## Conclusions

In conclusion, the current meta-analysis showed conspicuous risk factors affecting spinal fusion, including three patient-related risk factors (smoking, vitamin D deficiency, diabetes) and four surgery-related risk factors (without the use of BMP-2, allograft, CPS fixation, and posterolateral fusion). These findings may help clinicians strengthen awareness for early intervention in patients at high risk of developing fusion failure.

## Supporting information

**S1 Checklist. PRISMA checklist.**
(DOCX)

**S2 Checklist. PRISMA 2020 checklist.**
(DOCX)

**S1 Table. General search strategies for PubMed, Embase and Cochrane Library.**
(DOCX)

**S2 Table. Methodological quality score of the included studies based on the Newcastle—Ottawa Scale (NOS) tool.**
(DOCX)

**S3 Table. Grading evidence based on Egger's P value, sample size and heterogeneity.**
(DOCX)

**S4 Table. List of included and excluded studies.**
(DOCX)

**S5 Table. Fusion rates by study-level factors.**
(DOCX)

**S6 Table. Sensitivity analysis for significant and non-significant factors and class of evidence.**
(DOCX)

**S7 Table. Sensitivity analysis for fusion rate associated with patient-related factors and surgery-related factors.**
(DOCX)

**S1 Fig. Forest plot for pooled fusion rate.**
(TIF)

**S2 Fig. Funnel plots for meta-analysis of association between smoking and fusion rate.**
(TIF)

**S3 Fig. Funnel plots for meta-analysis of association between graft type (allograft vs. autograft) and fusion rate.**
(TIF)

**S4 Fig. Funnel plots for meta-analysis of association between number of fused levels (two vs. single) and fusion rate.**
(TIF)

**S5 Fig. Funnel plots for meta-analysis of association between without the use of BMP-2 (yes vs. no) and fusion rate.**
(TIF)

**S6 Fig. Funnel plots for meta-analysis of association between vitamin D deficiency and fusion rate.**
(TIF)

**S7 Fig. Funnel plots for meta-analysis of association between pedicle screw type (CPS vs. EPS) and fusion rate.**
(TIF)

**S8 Fig. Funnel plots for meta-analysis of association between diabetes and fusion rate.**
(TIF)

**S9 Fig. Funnel plots for meta-analysis of association between MIS and fusion rate.**
(TIF)

**S10 Fig. Funnel plots for meta-analysis of association between fusion column (posterolateral vs. lateral) and fusion rate.**
(TIF)

## Author Contributions

**Conceptualization:** Beijun Zhou, Mingjiang Luo.

**Data curation:** Shudong Yang, Beijun Zhou, Jiaxuan Mo, Ruidi He, Gaigai Yang, Mingjiang Luo, Zhihong Xiao.

**Formal analysis:** Kunbo Mei, Zhi Zeng, Yuwei Chen, Mingjiang Luo.

**Funding acquisition:** Zhihong Xiao.

**Methodology:** Shudong Yang, Yuwei Chen, Mingjiang Luo.

**Resources:** Beijun Zhou.

**Software:** Beijun Zhou, Jiaxuan Mo.

**Supervision:** Shudong Yang, Ruidi He, Zhihong Xiao.

**Validation:** Shudong Yang.

**Writing – original draft:** Mingjiang Luo, Siliang Tang.

**Writing – review & editing:** Siliang Tang.

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
