## [Decision Letter · Decision Letter 0]

20 Feb 2024

PONE-D-24-02124Risk Factors Affecting Spinal Fusion: A Meta-Analysis of 39 Cohort StudiesPLOS ONE

Dear Dr. Luo,

Thank you for submitting your manuscript to PLOS ONE. After careful consideration, we feel that it has merit but does not fully meet PLOS ONE’s publication criteria as it currently stands. Therefore, we invite you to submit a revised version of the manuscript that addresses the points raised during the review process.

We look forward to receiving your revised manuscript.

Kind regards,

Jae-Young Hong

Academic Editor

PLOS ONE

-https://link.springer.com/article/10.1007/s10143-023-02041-0?

https://journals.lww.com/international-journal-of-surgery/fulltext/2023/10000/risk_factors_of_epidural_hematoma_in_patients.30.aspx

In your revision ensure you cite all your sources (including your own works), and quote or rephrase any duplicated text outside the methods section. Further consideration is dependent on these concerns being addressed.

“This work was supported by the Health and Family Planning Commission Program of Hunan Province (no. 202204074707), the Health and Family Planning Commission Program of Wuhan City (no. WX18C29), and Natural Science Foundation of Hunan Province (no. 2022JJ30516).”

4. In this instance it seems there may be acceptable restrictions in place that prevent the public sharing of your minimal data. However, in line with our goal of ensuring long-term data availability to all interested researchers, PLOS’ Data Policy states that authors cannot be the sole named individuals responsible for ensuring data access (http://journals.plos.org/plosone/s/data-availability#loc-acceptable-data-sharing-methods).

5. Please ensure that you include a title page within your main document. You should list all authors and all affiliations as per our author instructions and clearly indicate the corresponding author.

6. Please include your tables as part of your main manuscript and remove the individual files. Please note that supplementary tables (should remain/ be uploaded) as separate "supporting information" files.

Reviewers' comments:

Reviewer's Responses to Questions

**Comments to the Author**

1. Is the manuscript technically sound, and do the data support the conclusions?

Reviewer #1: Yes

Reviewer #2: Partly

2. Has the statistical analysis been performed appropriately and rigorously? 

Reviewer #1: Yes

Reviewer #2: I Don't Know

3. Have the authors made all data underlying the findings in their manuscript fully available?

Reviewer #1: Yes

Reviewer #2: No

4. Is the manuscript presented in an intelligible fashion and written in standard English?

Reviewer #1: Yes

Reviewer #2: Yes

5. Review Comments to the Author

Reviewer #1: Thank you for submitting your work to PLOS ONE. The study aims to explore the potential risk factors regarding nonunion after spinal fusion surgery, leveraging a comprehensive meta-analysis approach.

Before considering this manuscript for publication, I recommend a major revision based on the following aspects.

1. Surgical variability and fusion rate analysis should be added. Your study predominantly focused on general risk factors affecting spinal fusion without considering the influence of surgical variability, such as the type of surgery or the site of surgery. Given that these variables can significantly impact fusion outcomes, it would enhance the manuscript's value to report on how different types of surgeries or fusion sites correlate with fusion rates. It offer a more nuanced understanding of risk factors.

2. Inclusion of forest plots in the main manuscript, not in supplement materials. While it is noted that forest plots were provided as supplementary figures, integrating these plots into the main figures of the manuscript would likely improve the accessibility and interpretability of your results for readers.

3. I also recommend adding tables that show baseline characteristics of the references, such as country, type and site of surgery, sample size, mean follow-up period, as well as included variables and significant variables for each reference.

Reviewer #2: 1. In statistics and other analyses

1) There is a need to explain the data extraction process in meta-analysis in more detail.

2) The method for calculating the effect size of data extracted from each study must be described. Then, the authors must explain in detail how authors integrated these effect sizes and performed statistical analysis.

2.From the results,

1) The authors mentioned MIS and the number of fused levels, and it seems that this should be described as a factor to be analyzed in the method. All factors to be analyzed must be introduced in the method, and the analysis details must be reported in the results.

2) The authors found that factors that inhibit spinal fusion, such as smoking, diabetes, vitamin D deficiency, graft type, pedicle screw type, and fusion column, were extracted from 39 studies, but the number of studies was small, with only more than 500 subjects or an I2 value of more than 0.5. Is this why it is classified as high or medium quality? I think we need to present objective standards for this.

6. PLOS authors have the option to publish the peer review history of their article (what does this mean?). If published, this will include your full peer review and any attached files.

Reviewer #1: No

Reviewer #2: No

---

## [Author Response · Author response to Decision Letter 0]

15 Apr 2024

Detailed Rebuttal Letter to Editor and Reviewer

Dear Prof. Hassan Zmerly, 

Thank you for forwarding the Reviewer comments to us. We have carefully gone through their feedback and have made the necessary changes to our manuscript. We appreciate the insightful feedback provided by the reviewers, as they have helped us improve the quality of our research. Please find below a detailed list of the changes made in response to the Editor and each of the Reviewer's concerns:

Editor’s comments: 

Response: We respond to each point of view put forward by academic editors and reviewers, and if necessary, we modify the content of the manuscript according to the suggestions of academic editors and reviewers (the changes are marked in red font). Finally, we will upload this letter as a separate file labeled 'Response to Reviewers'.

Response: All the revised contents in the manuscript are highlighted in red font. And we will upload this as a separate file labeled 'Revised Manuscript with Track Changes'.

Response: We separately uploaded a manuscript file without any markup, which was marked as "manuscript".

4) Response: Thank editor for reminding us that we do not need to change our financial disclosure.

Journal Requirements:

1)Please ensure that your manuscript meets PLOS ONE's style requirements, including those for file naming. The PLOS ONE style templates can be found at 

Response: We re-referenced the specified PLOS One style template to adjust the content format of our manuscript to ensure that our manuscript meets the style requirements of PLOS One. 

2) We noticed you have some minor occurrence of overlapping text with the following previous publication(s), which needs to be addressed:

-https://link.springer.com/article/10.1007/s10143-023-020410?

https://journals.lww.com/international-journal-of-

surgery/fulltext/2023/10000/risk_factors_of_epidural_hematoma_in_patients.30.aspx

In your revision ensure you cite all your sources (including your own works), and quote or rephrase any duplicated text outside the methods section. Further consideration is dependent on these concerns being addressed.

Response: We are very grateful to the journal team for their careful review of our article and pointed out that there is some duplication in our manuscript. In response to this important suggestion, we carefully reviewed all the contents of our article and revised the repeated parts.

3) Thank you for stating the following in the Funding Section of your manuscript:

“This work was supported by the Health and Family Planning Commission Program of Hunan Province (no. 202204074707), the Health and Family Planning Commission Program of Wuhan City (no. WX18C29), and Natural Science Foundation of Hunan Province (no. 2022JJ30516).”

Response: We re-examined the Funding section of the manuscript and deleted the description of the funding statement in the manuscript.

Accordingly, we have revised the funding section as follows: 

The authors received no specific funding for this work.

4) Before we proceed with your manuscript, please also provide non-author contact information (phone/email/hyperlink) for a data access committee, ethics committee, or other institutional body to which data requests may be sent. If no institutional body is available to respond to requests for your minimal data, please consider if there any institutional representatives who did not collaborate in the study, and are not listed as authors on the manuscript, who would be able to hold the data and respond to external

requests for data access? If so, please provide their contact information (i.e., email address). Please also provide details on how you will ensure persistent or long-term data storage and availability.

Response: All relevant data for this study can be obtained by corresponding author.

5) Please ensure that you include a title page within your main document. You should list all authors and all affiliations as per our author instructions and clearly indicate the corresponding author.

Response: We list all authors and all affiliations as per our author instructions and clearly indicate the corresponding author in the title page. 

6) Please include your tables as part of your main manuscript and remove the individual files. Please note that (should remain/ be uploaded) as separate "supporting information" files.

Response: We have placed the table in the appropriate place in the article and deleted the separate files. In addition, we also upload "supporting information" as a separate file.

7）Please include captions for your Supporting Information files at the end of your manuscript, and update any in-text citations to match accordingly. Please see our Supporting Information guidelines for more information: http://journals.plos.org/plosone/s/supporting-information.

Response: We added the title information of the supporting material at the end of the manuscript and updated the reference in the manuscript.

Reviewer #1: 

Comments to the Author

Thank you for submitting your work to PLOS ONE. The study aims to explore the potential risk factors regarding nonunion after spinal fusion surgery, leveraging a comprehensive meta-analysis approach. Before considering this manuscript for publication, I recommend a major revision based on the following aspects.

1) Surgical variability and fusion rate analysis should be added. Your study predominantly focused on general risk factors affecting spinal fusion without considering the influence of surgical variability, such as the type of surgery or the site of surgery. Given that these variables can significantly impact fusion outcomes, it would enhance the manuscript's value to report on how different types of surgeries or fusion sites correlate with fusion rates. It offers a more nuanced understanding of risk factors.

Response: The reviewers pointed out that we should increase the analysis of the influence of surgical variability (including surgical site and type of operation) on the fusion rate. We are very grateful to the reviewers for this important suggestion, and we very much agree with the reviewers' suggestion. Therefore, we re-extract the data from the included study, and explore the effects of different surgical sites and types of surgery on the fusion rate through stratified analysis, which will enrich our research content and results.

Accordingly, we have revised the result section as follows: 

In addition, in the stratified analysis of surgical sites and surgical methods, we found that the fusion rate of cervical surgery was 92% (95% CI, 89.7% to 94.3%, I2=73.7%, P < 0.001), which was Higher than the fusion rate of lumbar surgery. And in lumbar fusion surgery, the fusion rate of lateral approach was significantly higher than that of other approaches (Rate, 95.3%; 95% CI, 90.8% to 99.7%, I2=70.2%, P < 0.001).

(Page 16, Paragraph 1)

2) Inclusion of forest plots in the main manuscript, not in supplement materials. While it is noted that forest plots were provided as supplementary figures, integrating these plots into the main figures of the manuscript would likely improve the accessibility and interpretability of your results for readers.

Response: We are very grateful to the reviewers for their important suggestions for our study. The reviewer pointed out that we should put the forest plots in the manuscript to improve the reader's accessibility and interpretability of the results. We very much agree with the reviewer's point of view. In order to respond to the important suggestion of the reviewer, we included all the forest plots in the main manuscript. At the same time, the title of the picture is added where it is quoted in the manuscript.

3) I also recommend adding tables that show baseline characteristics of the references, such as country, type and site of surgery, sample size, mean follow-up period, as well as included variables and significant variables for each reference.

Response: The reviewers suggested that show baseline characteristics of the references, such as country, type and site of surgery, sample size, mean follow-up period, as well as included variables and significant variables for each reference. We believe that this suggestion of the reviewers is of great importance and is of great help to improve the quality of this study. For this reason, we have added information about the inclusion of country, type and site of surgery and sample size and mean follow-up period in Table 1.

Reviewer #2: 

Comments to the Author

1. In statistics and other analyses 

1) There is a need to explain the data extraction process in meta-analysis in more detail.

Response: We sincerely thank the reviewers for the recognition of our study and work. In addition, the reviewers pointed out that we should explain the data extraction process in Meta analysis in more detail. We also think that this is very necessary. To this end, we have added and modified the content about the data extraction process in the methodology section of the manuscript.

Accordingly, we have revised the methods section as follows: 

Two authors extracted data using a predesigned data extraction sheet. The specific extracted content was obtained by the relevant authors by reading the full text of the article and the contents of the table. When the data was incomplete or missing, we tried to contact the corresponding author of the article to obtain the relevant data. Differences between researchers in the process of extracting data were resolved through discussion or negotiation with third parties. 

(Page 6, Paragraph 8)

2) The method for calculating the effect size of data extracted from each study must be described. Then, the authors must explain in detail how authors integrated these effect sizes and performed statistical analysis.

Response: The reviewer pointed out that we should add the method of calculating the effect amount and the description of how to carry on the statistical analysis to the effect amount in the manuscript. We very much agree with and thank the reviewers for this proposal. In this regard, our data processing method is described in more detail in the statistical analysis part of the method.

Accordingly, we have revised the methods section as follows: 

The odds in each group were computed as p/(1-p) where p represents the proportion with exposure. The odds ratio (OR) was determined by dividing the odds in the fusion failure group by the odds in the comparator group. In the meta-analyses, study-specific log odds ratios were utilized as the outcome, and the aggregated estimates were then transformed into OR. If the OR > 1, it indicates a higher probability of fusion failure in the exposed group as opposed to the non-exposed group. 

(Page 8, Paragraph 1)

2.From the results, 

1) The authors mentioned MIS and the number of fused levels, and it seems that this should be described as a factor to be analyzed in the method. All factors to be analyzed must be introduced in the method, and the analysis details must be reported in the results.

Response: We are very grateful to the reviewers for their careful review of our research. Because of our mistake, we omitted the introduction of MIS and the number of fused levels in our method. For this reason, we revised the manuscript appropriately and re-added the content related to MIS and the number of fused levels.

Accordingly, we have revised the methods section as follows: 

All of our analyses were performed using Stata software (Stata version 16.0, College Station, Texas, USA). We analyzed the risk factors affecting spinal fusion, including patient-related factors (e.g., smoking, diabetes, and vitamin D deficiency). surgery-related risk factors (e.g., allograft, without the use of BMP-2, conventional pedicle screw (CPS) fixation, and posterolateral fusion, MIS, number of fused levels)

(Page 9, Paragraph 1)

2) The authors found that factors that inhibit spinal fusion, such as smoking, diabetes, vitamin D deficiency, graft type, pedicle screw type, and fusion column, were extracted from 39 studies, but the number of studies was small, with only more than 500 subjects or an I2 value of more than 0.5. Is this why it is classified as high or medium quality? I think we need to present objective standards for this.

Response: The reviewers questioned the evidence level classification basis of our manuscript on risk factors. First of all, I would like to thank the reviewers for their important suggestions on the basis of the classification of the level of evidence. It has to be admitted that there is no uniform standard on the basis of the classification of the level of evidence. Previously, we have referred to a variety of criteria for the classification of the level of evidence. In the end, the standard currently used is selected. Because this set of standards is most applicable to this study. In addition, the standard for the classification of the level of evidence is now used frequently, and the articles expressed in a large number of high-scoring journals are also the criteria for the classification of this set of levels of evidence.

Grading Evidence Based on Egger’s P value, Sample Size and Heterogeneity.

Related reference

(1) Luo M, Cao Q, Zhao Z, et al. Risk factors of epidural hematoma in patients undergoing spinal surgery: a meta-analysis of 29 cohort studies. Int J Surg. 2023;109(10):3147-3158. Published 2023 Oct 1. doi:10.1097/JS9.0000000000000538

(2) Mei Z, Wang Q, Zhang Y, et al. Risk Factors for Recurrence after anal fistula surgery: A meta-analysis. Int J Surg. 2019;69:153-164. doi:10.1016/j.ijsu.2019.08.003

---

## [Decision Letter · Decision Letter 1]

14 May 2024

Risk Factors Affecting Spinal Fusion: A Meta-Analysis of 39 Cohort Studies

PONE-D-24-02124R1

Dear Dr. Mingjiang Luo

We’re pleased to inform you that your manuscript has been judged scientifically suitable for publication and will be formally accepted for publication once it meets all outstanding technical requirements.

Kind regards,

Jae-Young Hong

Academic Editor

PLOS ONE

Additional Editor Comments (optional):

Authors well revised the manuscript. 

Reviewers' comments:

Reviewer's Responses to Questions

**Comments to the Author**

1. If the authors have adequately addressed your comments raised in a previous round of review and you feel that this manuscript is now acceptable for publication, you may indicate that here to bypass the “Comments to the Author” section, enter your conflict of interest statement in the “Confidential to Editor” section, and submit your "Accept" recommendation.

Reviewer #1: All comments have been addressed

Reviewer #2: (No Response)

2. Is the manuscript technically sound, and do the data support the conclusions?

Reviewer #1: Yes

Reviewer #2: Yes

3. Has the statistical analysis been performed appropriately and rigorously? 

Reviewer #1: Yes

Reviewer #2: Yes

4. Have the authors made all data underlying the findings in their manuscript fully available?

Reviewer #1: Yes

Reviewer #2: Yes

5. Is the manuscript presented in an intelligible fashion and written in standard English?

Reviewer #1: Yes

Reviewer #2: Yes

6. Review Comments to the Author

Reviewer #1: Thank you for your vigorous effort to improve the manuscript according to the reviewers' comments. This manuscript seems much improved and met the reviewers's expectations.

Reviewer #2: Reviewer’s comments pointed out in the first review have been revised to comply. However, there are too many duplicate tables, and need to be organized. So, I recommend making a few changes to this.

1) First, I think it would be a good idea to organize and merge Table 1 and S2 table into one.

2) Also, the S1 table is explained in the Search strategy, but it would be better to supplement the content here and eliminate the table.

3) The S4 Table is also shown in Figure 1, so there is no need to insert it.

4) I think it would be a good idea to organize Table 2 and S6 Table and combine them into one.

Also, Figures S2 to S10 do not specify what the explanation is.

7. PLOS authors have the option to publish the peer review history of their article (what does this mean?). If published, this will include your full peer review and any attached files.

Reviewer #1: No

Reviewer #2: No

---

## [Editor Report · Acceptance letter]

29 May 2024

PONE-D-24-02124R1 

PLOS ONE

Dear Dr. Luo, 

I'm pleased to inform you that your manuscript has been deemed suitable for publication in PLOS ONE. Congratulations! Your manuscript is now being handed over to our production team.

Kind regards, 

on behalf of

Professor Jae-Young Hong 

Academic Editor

PLOS ONE